# Effect of Varying Levels of Chromium Propionate on Growth Performance and Blood Biochemistry of Broilers

**DOI:** 10.3390/ani9110935

**Published:** 2019-11-07

**Authors:** Muhammad Arif, Imtiaz Hussain, Muhammad A. Mahmood, Mohamed E. Abd El-Hack, Ayman A. Swelum, Mahmoud Alagawany, Ahmed H. Mahmoud, Hossam Ebaid, Ahmed Komany

**Affiliations:** 1Department of Animal Sciences, College of Agriculture, University of Sargodha, Sargodha 40100, Pakistan; 2Department of Poultry, Faculty of Agriculture, Zagazig University, Zagazig 44511, Egypt; 3Department of Theriogenology, Faculty of Veterinary Medicine, Zagazig University, Zagazig 44511, Egypt; 4Department of Zoology, College of Sciences, King Saud University, Riyadh 11451, Saudi Arabia

**Keywords:** chromium propionate, growth, carcass, immunity, blood, broilers

## Abstract

**Simple Summary:**

The current study evaluated the effects of dietary chromium propionate supplementation on growth performance and blood biochemistry of broilers. Results showed that chromium propionate has improved weight gain and feed conversion ratio (FCR) of broilers. Also, meat to fat ratio improved and lean meat may be produced. Based on the study results, the recommended level of chromium propionate supplementation to broiler diet for better performance and weight gain is 400 ppb.

**Abstract:**

The objective of this work was to evaluate the effects of varying levels of chromium propionate on blood biochemistry and growth performance of broilers (1–35 days). Five diets were formulated by using chromium propionate with inclusion levels of 0, 200, 400, 800 and 1600 ppb. A total of 300 broilers were divided into 5 groups with 6 replicates of 10 birds in each under completely randomized design. The starter feed intake remained unaffected (*p* > 0.05) whereas finisher and overall feed intake was different (*p* < 0.05) among different experimental groups. Feed conversion ratio and weight gain in starter, finisher and overall improved significantly (*p* < 0.05) with the increasing levels of chromium propionate. Blood glucose was decreased (*p* < 0.05) with increasing dietary chromium level. Chromium supplementation did not affect antibodies titers against NDV and AIV-H9. Neither live, hilal, after skin removal, eviscerated, chest weight and legs with shanks weight nor liver and heart weights were affected (*p* > 0.05) while gizzard weight reduced significantly (*p* < 0.05) due to supplementation of chromium. On the basis of results, it may be concluded that chromium propionate supplementation improved weight gain and FCR and reduced blood glucose. However, better performance and weight gain may be achieved if chromium propionate is added at the rate of 400 ppb in broiler diets.

## 1. Introduction

Feed additives have gained great popularity for their several benefits in humans and animals [1,2]. This research focuses on the use of chromium as a feed additive in poultry. The importance of chromium in animal nutrition is well-known for more than 40 years as it improves tissue sensitivity of insulin receptors, resulting in improved uptake of glucose by the cells [3]. It is considered biologically active and essential for animals and human beings with a difference in metabolic activities inside the body. So, it is used in diets of poultry and animals to improve performance and productivity [4]. Elemental chromium extraction can be made by dissolving natural states of chromium i.e., chromium nitrate, chromium acetate and chromium oxide in acids such as hydrochloric acid and sulfuric acid. However, it can also be obtained by reducing hexavalent chromium using cytochrome c7, which are proteins with heme that acts as a cofactor [4].

Trivalent Cr^+3^ was considered a nutrient in the late 1950s. After that time, many research studies were conducted to evaluate its beneficial effects in biological systems [5]. Its major contribution is to improve insulin sensitivity that support carbohydrates, fat and protein metabolism. In the United States, Cr^+3^ accepted as an essential nutrient for use in human whereas European Food Safety Authority (EFSA) described in 2014 that available proofs and data are not sufficient to recognizes Cr^+3^ as essential nutrient [6]. However, there are many countries that accepted Cr^+3^ as a nutrient such as Australia, New Zealand, Japan etc. Hexavalent Cr^+6^ is toxic and carcinogenic when inhaled or ingested but trivalent Cr^+3^ may also possess hypothetical risk when ingested above the threshold quantities whereas normal metabolism tries to prevent its toxicity [7]. However, Cr^+3^ are usually found in small quantities in few raw materials that are usually used in diets of broilers including grains and oilseed meals [8]. Moreover, Cr^+3^ is available in variety of foods such as egg yolks, meat products, whole cereal grains and their bran products, coffee beans and nuts, broccoli, brewery yeast, wines and beers [9]. Chromium contents in the food materials vary because of varying mineral levels in the soil, type of plant, season and processing. Moreover, stainless steel is coated with chromium plating and it can be leach out during cooking of foods [7,10].

Chromium’s role in carbohydrates, lipids and protein metabolism is well-documented. Glucose tolerance factor (GTF) in body which is activated by chromium is responsible to make metabolic activity of insulin more effective. Chromium is also essential for chromodulin, a cofactor of certain enzymes i.e., tyrosine kinase etc. which are required for proteins stabilization and formation of nucleic acids [11]. Physiological traits of broilers with supplemental chromium in forms of Cr-chloride, Cr-yeast and Cr-picolinate in feed reduced blood glucose concentrations [12]. Chromium supplementation can decrease carcass fat percentage and also reduced plasma cholesterol level in broilers [13]. Total cholesterol levels of breast meat and thigh meat decreased in experimental units supplemented with organic chromium at levels of 200 ppb and 400 ppb whereas triglycerides also reduced [14].

It has been also established that corticosterones levels were enhanced against stress factors that effected insulin by negative feedback and reduction of insulin leads to higher blood glucose concentrations. Supplemental chromium in diets of broilers enhanced insulin sensitivity and increased uptake of glucose to cells was observed. It has been published that supplemental chromium has elevated total serum protein concentrations i.e., albumin and insulin whereas reduction in corticosterone and cholesterol concentrations in blood were also observed [14]. In broilers, supplemental trivalent organic chromium enhanced growth, breast meat yield and carcass quality [15]. Liver weight relative with body weight percentage of birds has also showed improvements by supplemental chromium at inclusion of 800 ppb of the diet [16]. Quantitative and qualitative traits of meat had also been confirmed by the use of chromium in various studies [17]. Chromium supplementation in broilers has also reduced low density lipoproteins (LDL), triglycerides while increased high density lipoproteins (HDL). Increased excretion of chromium is observed in various types of stress conditions in many species that induce insulin resistance [17].

Dietary supplementation of chromium enhanced immune functions of chickens vaccinated with Avian Influenza Virus (AIV), and the chromium chloride was more effective than the chromium picolinate in improving the lymphoid organs weight, however, the chromium picolinate was more effective than the chromium chloride in improving the blood antibody titer against AI virus [18]. The data concerning the use of chromium propionate on antibody titers against Newcastle Disease Virus (NDV) and Avian Influenza Virus-Type 9 (AIV-H9) of broilers are scanty, and the previous studies on the use of chromium propionate in broiler rations resulted in contradictory conclusions. Therefore, the current research study has been designed to evaluate the dietary effects of chromium propionate on growth performance, blood biochemistry of broilers (i.e., glucose, liver enzymes alanine aminotransferase (ALT), aspartate aminotransferase (AST), alkaline phosphatase (ALP), cholesterol, LDL, HDL and triglycerides) and antibody titers against NDV and AIV-H9.

## 2. Materials and Methods

This study was conducted to evaluate the effects of dietary chromium propionate on growth performance and blood biochemistry in broilers at Poultry Research Center, College of Agriculture, University of Sargodha, Sargodha. All experimental procedures of the above-mentioned study were performed according to the Local Experimental Animal Care Committee and approved by the ethics of the institutional committee of College of Agriculture, University of Sargodha, Sargodha (SARU-0021-2019).

### 2.1. Birds, Housing and Management

For the experiment, 300 male ROSS-308-day-old broiler chicks were purchased and weighed initially and then randomly divided into the 5 experimental groups in arrangement under complete randomized design. Each group was replicated 6 times with 10 birds in each replicate. The total duration of the trial was 35 days. Experimental shed was cleaned thoroughly by scrubbing, dusting and washing with water. After cleaning the shed, it was white washed in which phenol was mixed as disinfectant and house was sprayed with mixture of formalin and water at 1:10 ratio with pump sprayer. Fumigation was also made before arrival of chicks.

Feeders and drinkers were washed with potassium permanganate (KMnO_4_) solution and dried up in sunlight to ensure complete disinfection. Chicks in each replicate were placed in separate pens with the same managerial conditions. Birds were housed in floor litter system and rice husks were used as bedding. Raking of bedding material was performed routinely. Biosecurity measures were followed strictly and trial was performed under complete hygienic conditions. Cleaning and washing of drinkers were performed routinely. Lightening program was followed by ROSS-308 management guide (2018). Shed temperature was adjusted to 95 °F for 1st week of age and afterwards, it was reduced by 5 °F in every week till reached to 75 °F. Afterwards, it was maintained throughout flock.

### 2.2. Experimental Diets

Chicks in five treatments were fed two mixtures of diets. The National Research Council (NRC) guidelines were followed to formulate the basal diets. These were divided into 2 phases: 1st phase was broiler starter from (1st–21st day) and 2nd phase was broiler finisher (22nd–35th day). Supplementation of chromium propionate with dietary levels of 0, 200, 400, 800 and 1600 ppb of feed were used to make five experimental diets C0, C1, C2, C3 and C4 respectively.

Chromium propionate i.e., KemTrace Cr (0.4%—4 mg of Cr/g of product) from manufacturer Kemin Industries, Inc., Des Moines, IA, USA. was purchased from local market and mixed with the experimental diets. Broiler birds were offered water and feed at ad-libitum. The ingredients of starter diet and finisher diet along with their composition have been shown in given Table 1.

### 2.3. Growth Performance

Feed intake was observed from all replicates of each dietary treatment. It was measured daily and weekly basis. Initial weight of day-old chick and weekly body weight were recorded for all the replicates. On the basis of feed intake and the body weight; feed conversion ratio was determined for all the replicates on weekly basis.

### 2.4. Carcass Evaluation

On the 35th day of age, randomly two birds were selected from each replicate for evaluation of carcass and visceral organs. Prior to slaughter, feed was withheld for six hours to emptying of gastrointestinal tract. Effects on carcass parameters and visceral organs were measured in terms of live weight, hilal weight, after skin removal, eviscerated weight, chest weight and legs with shanks weight. Liver, heart and gizzard weight were also measured. The carcass parameters and visceral organs weighing were performed on fresh basis.

### 2.5. Blood Biochemistry or Serum Metabolites

Blood samples from two birds of each replicate were collected on 35th day from wing vein in gel tube (SPS *tubes*/”*yellow tops*”) that contains a special type of *gel* that separates blood cells from serum and causing blood to clot quickly. Centrifugation of blood samples was made at 3000 RPM for 15 min. These serum collection tubes were kept in a deep refrigerator for biochemical analysis including glucose, liver enzymes i.e., ALT, AST and ALP. Total cholesterol, LDL, HDL and triglycerides were also measured according to the instructions of specific commercial kits.

### 2.6. Serology for Newcastle Disease Virus (NDV) and Avian influenza Virus H9 (AIV-H9)

Serum samples were also used for Hemagglutination inhibition (HI) test against NDV and AIV-H9. These tests were performed by using standard protocol described for HI titers [19].

### 2.7. Statistical Analysis

Statistical interpretation of the data collected from all parameters of this research study was performed by analysis of various techniques under Completely Randomized Design [20]. Means of all parameters were separated by using Tukey’s test.

## 3. Results

Starter feed intake remained unaffected (*p* > 0.05) whereas finisher and overall feed intake was different among different experimental groups (Table 2). Lowest finisher and overall feed intake were observed in group C4. Weight gain in starter, finisher and overall improved (*p* < 0.05) significantly among the different treatment groups. A linear trend in starter weight and quadratic trend in finisher and overall weight was observed in experimental groups. The lowest value of weight gain was observed in C4 supplemented group but the highest value was observed in C2 group. Regarding feed conversion ratio, a quadratic trend in starter and overall FCR was observed but finisher FCR showed a linear trend in chromium supplemental groups. The lowest value of FCR was observed in C2 but highest FCR was observed in C4 (Table 2).

Serum concentration of the lipid profile (LDL, HDL, triglycerides, and cholesterol) and AST, ALT and ALP were not significantly affected by chromium supplementation. Cr-propionate supplementation decreased (linear and cubic effect) serum glucose in comparison with the control group (*p* > 0.05), but did not affect liver enzymes (AST and ALT) and ALP (Table 3).

Antibodies titers against NDV and AIV-H9 were remained unaffected among the different experimental groups with increasing inclusion levels of Cr-propionate in broilers (Table 4). There were no significant (*p* > 0.05) differences in live weight, hilal weight, after skin removal weight, eviscerated weight, chest weight, legs with shanks weight, liver heart and gizzard weight due to Cr-propionate supplementation (Table 5).

## 4. Discussion

The lowest values of finisher and overall feed intake were observed in C4 group. Unaltered feed intake in starter phase of this study was in line with the results of Eze et al. [21] who observed no significant effect of supplemental chromium on feed intake in early weeks. Finisher and overall feed intake affected by increasing dose of chromium is in line with few researchers who expressed similar results that feed intake increases with increasing dose of chromium in different forms of chromium (Cr-picolinate, Cr-nicotinic acid, Cr-histidinate and Cr-propionate) under heat stressed broilers [22,23]. Contrary to findings of the present study, a research study observed no significant improvement in feed intake with supplemental chromium in broilers diet [24]. Some other researchers reported that supplemental Cr^+3^ in organic and inorganic forms did not affect feed intake [25,26]. Different sources of chromium basal diets and levels of chromium in broilers may contribute to variations of results.

Regarding body weight, similar to current findings, a significant increase in body weight gain was observed with Cr-yeast at 150 and 300 ppb [27]. Findings of the current study were also supported by some scientists who reported that dietary supplemental chromium as Cr-picolinate, Cr-methionine, Cr-chloride, Cr-yeast, Cr-propionate sources at different inclusion levels increased in weight gain under heat stress condition [25,28]. Supplemental Cr-yeast increases superoxide dismutase activity which reduces the oxidative stress and lipids peroxidation that ultimately reduces MDA levels which is good marker for stress [29]. It is well documented that chromium is essential for proper insulin functioning and also required for normal protein, fat and carbohydrate metabolism which is acknowledged as a dietary supplement in humans as well [30]. Moreover, greater uptake of glucose to the muscle and adipose tissues represents anabolism which increases serum growth factors concentrations i.e., insulin growth factor-1 (IGF-1) that increases the protein assembling in broilers [31]. Contrary to current study, supplemental chromium in the form of Cr-picolinate with the dietary doses of 0, 200, 400, 800 ppb did not affect the body weight gain [13]. Few other scientists also disagreed with present results and they revealed that chromium additions in form of Cr-picolinate and some other organic sources at different dose rates may not improve weight gain [24,32]. These variations in reports might be because of differences in types of chromium sources along with levels in basal diets and experimental methods used [17].

Regarding feed conversion ratio, decreased FCR in starter, finisher and overall phases were observed, which is in line with the previously reported study in which increasing dose rate of supplemental chromium from 0 to 3200 ppb in the form of Cr-propionate had reduced FCR in broilers [26]. The findings of the current study were also supported by some other researchers who observed reduced FCR with dietary inclusion of chromium at variable dose rates of organic and inorganic sources [28,33]. Reduction in FCR might be attributed to the maximum utilization of glucose from blood. According to a study, chromium supplementation in humans and animals showed increased insulin activity resulting in more glucose absorption and amino acids utilization to produce energy, muscles development and fat conversion [34]. Opposite to the current study, a researcher found that dietary supplementation of Cr^+3^ with the doses of 0, 200, 400 ppb in the form of Cr-picolinate did not affect the feed conversion ratio in broilers [35]. Some other studies also reported unaffected FCR with dietary inclusion of Cr^+3^ at different levels of Cr-yeast and inorganic sources [21,24,36]. This might be due to higher levels of chromium in basal diets and less stress to birds can increase the level of Cr^+3^ above the threshold that may cause negative impact on feed intake of the birds and also posses’ hypothetical risk to animals.

In the present study, dietary chromium supplementation did not affect serum lipid profile. Unchanged values of LDL, HDL, triglycerides and cholesterol were in line with the outcomes of Kani [37] who observed no difference of chromium on serum LDL, HDL and cholesterol. Few other scientists also supported the current findings who found no effect on chromium either organic or inorganic forms on serums LDL, HDL and cholesterol [38]. Elevated levels of blood corticosterone, glucose, cholesterol and triglycerides were observed in response to the stress that occasionally occurs in broilers. Serum values of cholesterol, HDL and LDL were not significantly affected by the addition of chromium to broilers diet [39]. Contrary to current findings, some previous studies observed that serum LDL, and cholesterol reduced whereas HDL increased with the addition of various organic and inorganic sources of Cr-methionine, Cr-yeast, Cr-picoloinate and Cr-chloride [14,15,40]. On the other hand, supplemental chromium increased triglycerides at age of 42 days in comparison with control [36]. Few other researchers also agreed with the previous results and they found that supplementation of organic sources of chromium in form of Cr-picolinate, Cr-methionine at varying dose rates increased serum triglycerides [41,42]. Contrary to present findings, a few previous studies observed that dietary supplementation of Cr-nicotinic acid, Cr-propionate with the dose 1500 ppb in diet of broilers under heat stress condition reduced level of triglycerides [43,44]. Variations in effects might be related to different sources of chromium used and dietary inclusion levels, stress condition and disease incidence [17].

In the current study, a reduction in blood level of glucose is in line with a previous study in which blood glucose levels decreased with supplemental Cr-propionate at varying levels in broiler diets [10,32]. Some other scientists also supported the observations of this current study and found that chromium in different forms i.e., Cr-picolinate, Cr-methionine, Cr-propionate, Cr-yeast and Cr-chloride at different dose rates decreases blood glucose [45,46]. This might be due to important metabolic function of chromium is to accelerate the activity of insulin by presence of organometallic compound which is known as GTF [47]. Chromium is also considered as an essential component of accelerating enzymes for maintaining the stability protein and nucleic acids [11]. This indicates enhanced activity of glucose tolerance factor for insulin utilization in broilers. At low concentrations of insulin, glucose is stored in adipose tissue in fat or glycogen form [48]. The ability of pancreatic insulin to regulate the fat metabolism and blood levels of glucose is dependent on the binding of insulin to receptors found in many peripheral tissues like adipocytes, muscle and liver. This process may involve the ability of chromium to regulate reactions of dephosphorylation and phosphorylation which turn insulin action on and off [49]. Contrary to current findings, a previous study also reported that supplemental chromium did not affect blood glucose [50]. This might be due to variation in basal levels of chromium and source used in this experiment. The findings of liver enzymes i.e., ALT, AST and ALP in this study agree with the findings of previous study in which researchers found that supplemental chromium in broilers did not affect liver enzymes [31]. Another study also supported the results of present study and found no difference of liver enzymes by chromium additions to broilers [51].

Regarding antibodies titers against NDV and AIV-H9, our findings agree with the outcomes of a previous research in which antibodies titer against NDV was similar at different dietary inclusion of chromium propionate [21]. Contrary to findings of present study, a research study found increased antibodies titers against NDV and AIV when Cr-picolinate was supplemented with the dose 1500 ppb in diet of broilers under heat stress condition [52]. Some of the other scientists found opposite results to present findings and documented increase in antibodies titers against NDV and AIV with supplemental Cr-methionine, Cr-chloride and Cr-propionate [18,53]. But these studies were conducted in heat stress conditions in which supplemental chromium may improve immunity due to increased Cr^+3^ losses of body.

Outcomes of this study match with a research study in which no significant difference was noticed in carcass characteristics of the birds with Cr-propionate supplementation [32]. Findings of this experiment were also endorsed by some other scientists who observed non-significant improvements in carcass traits [21,24]. Similar outcomes were also observed in a previous study and found no changes were observed in liver and heart weight in broilers [36]. Contrary to current study, Cr-methionine supplementation increased carcass yield in broilers during heat stress period [25]. Findings of few other researchers were also not in agreement with present study, describing that dietary supplementation of chromium in Cr-picolinate, Cr-nanocomposite, Cr-nicotinic, Cr-methionine, Cr-chloride forms in broilers increased the carcass and breast meat yield [26,54]. In these studies, higher relative percentages of meat yield to fat in carcass might increase the meat contents [55]. Reduction in abdominal fat was observed because of absence in process of lipogenesis as most of the glucose is uptaken by the cells increased sensitivity of insulin and no extra glucose in blood is left for lipids formation in broilers fed chromium [16]. Contrary to present study, it was also reported that percentage of liver weight and heart weight were reduced relatively to body weight offered with chromium in diets of broilers among the experimental groups [16].

On the basis of current outcomes, it may be concluded that Cr-propionate supplementation improved weight gain and FCR as well as reduced blood glucose. However, better performance and weight gain may be achieved if chromium propionate is added at the rate of 400 ppb in diet of broilers.

## Figures and Tables

**Table 1 animals-09-00935-t001:** Basal diets of broiler starter and broiler finisher.

Ingredients	Starter Usage (%)	Finisher Usage (%)
Corn	59.60	66.03
Soybean Meal	22.00	25.80
Canola Meal	12.00	0.00
Sunflower Meal	3.43	4.20
Corn Gluten 60%	0.00	1.00
Limestone	0.96	0.93
Dicalcium phosphate (DCP)	0.55	0.47
l-Lysine Sulphate	0.50	0.57
dl-Methionine	0.20	0.23
Premix *	0.20	0.20
Salt	0.20	0.20
Sodium bicarbonate	0.15	0.15
l-Threonin	0.08	0.09
l-Isoleucin 98%	0.13	0.13
Proximate analysis		
Dry matter	89.0	90.0
Crude protein	20.7	19.8
Ether extract	2.5	2.6
Crude fiber	4.0	3.5
Ash	5.0	4.3

* = Phytase (10,000FTU) 100 gm/ton = 0.01%, maduramycin 1% (500 gm/t) = 0.05%, betaine HCL = 0.05%, flavomycin = 0.02%, vitamins; minerals premix composition is given table.

**Table 2 animals-09-00935-t002:** Effect of chromium propionate in broiler growth performance.

Items	Treatments					SEM	Significance		
	C0	C1	C2	C3	C4		Linear	Quadratic	Cubic
Feed intake (g)									
0–21 days	1299.8	1289.1	1301.6	1279.4	1301.9	11.499	NS	NS	NS
22–35 days	2038.7 ^a^	1983.1 ^b^	2030.5 ^a^	2062.4 ^a^	1964.3 ^b^	12.354	NS	NS	*
0–35 days	3338.5 ^a^	3272.2 ^b^	3332.1 ^a^	3341.8 ^a^	3266.2 ^b^	14.104	NS	NS	*
Weight gain (g)									
0–21 days	977.8 ^c^	987.8 ^b,c^	1025.4 ^a^	995.4 ^b^	1003.5 ^b,c^	5.57	*	*	NS
22–35 days	1155.7 ^b^	1158.2 ^b^	1178.9 ^a^	1156.5 ^b^	1080.8 ^c^	7.81	NS	*	NS
0–35 days	2133.5 ^b^	2146.0 ^b^	2204.3 ^a^	2151.9 ^b^	2084.3 ^c^	12.91	NS	*	NS
Feed conversion ratio (g/g)									
0–21 days	1.3293 ^a^	1.3050 ^a,b^	1.2695 ^b^	1.2854 ^b^	1.2973 ^a,b^	0.0124	NS	*	NS
22–35 days	1.7647 ^b^	1.7125 ^c^	1.7231 ^c^	1.7833 ^a,b^	1.8175 ^a^	0.014	*	*	NS
0–35 days	1.5650 ^a^	1.5248 ^b^	1.5117 ^b^	1.5530 ^a^	1.5670 ^a^	8.44	NS	*	NS

C0, C1, C2, C3, C4 indicate supplementation of chromium propionate in the diets at the rate of 0, 200, 400, 800, 1600 ppb of feed, respectively. * = significant (*p* < 0.05). NS = non-significant (*p* > 0.05). ^a b c^ within a row, means sharing different superscripts differ significantly (*p* < 0.05).

**Table 3 animals-09-00935-t003:** Effect of chromium propionate on blood metabolites of broilers at slaughtering.

Items (mg/dL)	Treatments					SEM	Significance		
	C0	C1	C2	C3	C4		Linear	Quadratic	Cubic
LDL	56.0	46.167	39.833	50.333	50.667	5.778	NS	NS	NS
HDL	57.167	52.500	56.833	56.667	56.00	4.205	NS	NS	NS
Cholesterol	129.17	112.17	117.33	127.17	127.33	8.799	NS	NS	NS
Triglycerides	79.17	66.67	107.50	100.83	102.50	11.964	NS	NS	NS
Glucose (mg/dL)	248.17 ^a^	219.00 ^b^	213.67 ^b^	226.00 ^b^	211.33 ^b^	6.168	*	NS	*
AST (IU/L)	70.833	69.333	73.167	70.500	69.333	3.182	NS	NS	NS
ALT (IU/L)	3.833	4.000	3.833	4.000	4.166	0.469	NS	NS	NS
ALP (IU/L)	5.166	5.666	5.500	4.833	5.833	0.5077	NS	NS	NS

C0, C1, C2, C3, C4 indicate supplementation of chromium propionate in the diets at the rate of 0, 200, 400, 800, 1600 ppb of feed respectively. * = significant (*p* < 0.05). NS = non-significant (*p* > 0.05). ^a b^ within a row, means sharing different superscripts differ significantly (*p* < 0.05). LDL = low density lipoproteins, HDL = high density lipoproteins, AST: Aspartate Aminotransferase, ALT: Alanine Aminotransferase, ALP: Alkaline Phosphatase.

**Table 4 animals-09-00935-t004:** Effect of chromium propionate on the immune response of the broilers.

Items	Treatments					SEM	Significance		
	C0	C1	C2	C3	C4		Linear	Quadratic	Cubic
NDV (HI titers)	5.00	3.66	3.33	5.66	5.33	1.382	NS	NS	NS
AIV-H9 (HI titers)	5.33	4.00	3.66	5.33	3.00	0.966	NS	NS	NS

C0, C1, C2, C3, C4 indicate supplementation of chromium propionate in the diets at the rate of 0, 200, 400, 800, 1600 ppb of feed respectively. NS = non-significant (*p* > 0.05). NDV = Newcastle disease virus, AIV = avian influenza virus. HI test was performed for NDV; AIV-H9 and titers were calculated in GMT.

**Table 5 animals-09-00935-t005:** Effect of chromium propionate on carcass characteristics.

Items (g)	Treatments					SEM	Significance		
	C0	C1	C2	C3	C4		Linear	Quadratic	Cubic
Live weight	2156.8	2284.8	2239.5	2203.3	2095.5	117.15	NS	NS	NS
Hilal weight	2075.2	2207.8	2179.0	2123.2	2025.7	112.30	NS	NS	NS
After skin removal	1761.2	1931.7	1911.8	1847.7	1772.3	101.67	NS	NS	NS
Eviscerated weight	1440.8	1631.3	1603.8	1552.6	1477.2	88.123	NS	NS	NS
Chest weight	622.67	708.33	671.67	668.50	648.33	39.616	NS	NS	NS
Legs with shanks weight	552.67	571.83	571.50	552.50	549.33	30.239	NS	NS	NS
Liver	54.667	46.500	50.833	48.167	48.000	4.186	NS	NS	NS
Heart	12.833	12.000	12.000	11.500	12.333	0.830	NS	NS	NS
Gizzard	85.000	75.500	77.333	72.500	72.667	4.175	NS	NS	NS

C0, C1, C2, C3, C4 indicate supplementation of chromium propionate in the diets at the rate of 0, 200, 400, 800, 1600 ppb of feed respectively. NS = non-significant (*p* > 0.05).

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
