# Peer review of "Effect of Varying Levels of Chromium Propionate on Growth Performance and Blood Biochemistry of Broilers"

_animals, 2019, doi:10.3390/ani9110935_

Round 1
Reviewer 1 Report
Dear Authors,
thank you for submitting your research results.
Major improvement:
It was nice to read a paper with many references to other studies. Unfortunately, I do not see the need and outcome of your study. Could you please elucidate the aim of your study and the proper outcome (compared to the other studies)?
Minor improvements:
Furthermore, your manuscript needs minor improvements in content, presentation and, spelling as well as formatting.
Line 17: Please add “and” before “blood chemistry”.
Line 30: For better understanding, you might add “increased [chromium treatments]”
Line 31: I do not understand the use of “also” in this sentence and ask you to delete it.
Line 32: Please delete “(p>0.05)”
Line 33: Please format Cr+3 like in the main text.
Line 49 until line 57: References are missing and must be added.
Line 60 until line 63: Reference are missing and must be added.
Line 67: Why “,” after e.r.?
Line 77: Delete “levels increased” due to the fact, that “elevated” includes this
Line 103: Please do not only use the chemical formula.
Line 112: Please do not use abbreviations without explanation.
Line 119: Please discuss the possible effect of the other ingredients on your research aim.
Line 112 f: repetition
Line 125 ff: Was the weight measured individually or by group? Please clarify this.
Line 149: Please do not use abbreviations without explanation.
Line 161: First, “best” is a judgement and should not be in the result part. Second, you have the same sentence in the discussion part.
Line 163 ff / Table 2: Please state the deviations.
Table 2: 1301.9 + 3266.1 is 3266.2
Table 2: The line “Weigh gain 0-35 d” is nonsense. Are you measuring the overall gain from starting 0? Then the previous line (22-35 d) is wrong… Please be precise and clarify it.
Line 176: Please add the explanations for AST, ALT.
Line 194 ff: Maybe this is just a different style, but I recommend not to repeat the results in detail (which you are describing in the result part). You might delete many sentences in the discussion part.
Line 228: I do not agree with your sentence. Comparing the starter phase and the finisher phase, the feed conversion ratio the increased. But comparing the different groups, you can not state a linear increase. You are not precise here.
Line 230: Please explain your assumption of “best”.
Line 264: I recommend to change the formatting here.
Line 277 f: Please expand the explanation.
References: Please check the formatting (numbers are doubled etc.).
I really hope that you can appreciate the effort reviewing manuscripts.
Kind regards
Author Response
Major improvement:
It was nice to read a paper with many references to other studies. Unfortunately, I do not see the need and outcome of your study. Could you please elucidate the aim of your study and the proper outcome (compared to the other studies)?
Thank you very much for your great effort and valuable time you spent in reviewing our paper and for giving us the chance to clarify our work. We tried to improve some parts of introduction and discussion to give a better understanding of our point of view and we wish these changes would improve our paper upon your helpful suggestions. The aim of the study has been added.
Minor improvements:
Furthermore, your manuscript needs minor improvements in content, presentation and, spelling as well as formatting.
Thank you very much for such valuable comment. We already revised the linguistic and grammar of the manuscript and we wish it would be now better.
Line 17: Please add “and” before “blood chemistry”. Done
Line 30: For better understanding, you might add “increased [chromium treatments]” Done
Line 31: I do not understand the use of “also” in this sentence and ask you to delete it. Deleted
Line 32: Please delete “(p>0.05)” Deleted
Line 33: Please format Cr+3 like in the main text. Done
Line 49 until line 57: References are missing and must be added. Done
Line 60 until line 63: Reference are missing and must be added. Done
Line 67: Why “,” after e.r.? Done
Line 77: Delete “levels increased” due to the fact, that “elevated” includes this Done
Line 103: Please do not only use the chemical formula. Done
Line 112: Please do not use abbreviations without explanation. Done
Line 119: Please discuss the possible effect of the other ingredients on your research aim.
Done
Line 112 f: repetition. Done
Line 125 ff: Was the weight measured individually or by group? Please clarify this. Done-By replicate
Line 149: Please do not use abbreviations without explanation. Done
Line 161: First, “best” is a judgement and should not be in the result part. Second, you have the same sentence in the discussion part. Done
Line 163 ff / Table 2: Please state the deviations. We used the mean of standard errors
Table 2: 1301.9 + 3266.1 is 3266.2. Yes you are right…thanks a lot for your observation
Table 2: The line “Weigh gain 0-35 d” is nonsense. Are you measuring the overall gain from starting 0? Then the previous line (22-35 d) is wrong… Please be precise and clarify it.
You are right…Body weight gain has been corrected…thank you very much for your strong observations.
Line 176: Please add the explanations for AST, ALT. Done
Line 194 ff: Maybe this is just a different style, but I recommend not to repeat the results in detail (which you are describing in the result part). You might delete many sentences in the discussion part.
Done
Line 228: I do not agree with your sentence. Comparing the starter phase and the finisher phase, the feed conversion ratio the increased. But comparing the different groups, you can not state a linear increase. You are not precise here. Corrected
Line 230: Please explain your assumption of “best”. Done
Line 264: I recommend to change the formatting here. Done
Line 277 f: Please expand the explanation. Done
References: Please check the formatting (numbers are doubled etc.).
Thank you very much for this accurate revision, we considered the journal style in this reference and all other references
I really hope that you can appreciate the effort reviewing manuscripts.
Thank you very much for this accurate revision, we considered the journal style in this reference and all other references
Reviewer 2 Report
The content presented in this manuscript titled 'Effect of varying levels of chromium propionate on growth performance and blood biochemistry of broilers' is supported by the research data presented.
The manuscript could use a revision for grammatical errors and typos.
My question to authors is this is a typical blood profile work done in many studies previously, so why didn't the authors try anything novel to investigate the mode of actions of Chromium such as digestive enzyme activity, nutrient transporter expression, etc.?
There are lots of literature that have already looked into the parameters as the authors mentioned in the manuscript. Its a known knowledge that Chromium increases glucose uptake and may reduce triglyceride levels. So it would have been good to look at body composition of these birds to validate those rather than repeating the same again.
Author Response
Comments and Suggestions for Authors
The content presented in this manuscript titled 'Effect of varying levels of chromium propionate on growth performance and blood biochemistry of broilers' is supported by the research data presented.
The manuscript could use a revision for grammatical errors and typos.
Thank you very much for such valuable comment. We already revised the linguistic and grammar of the manuscript and we wish it would be now better.
My question to authors is this is a typical blood profile work done in many studies previously, so why didn't the authors try anything novel to investigate the mode of actions of Chromium such as digestive enzyme activity, nutrient transporter expression, etc.?
Thank you very much for these valuable comments and observations. To be honest, the lack of facilities and financial support for scientific work in our country stop us from performing the all required mechanisms and parameters such as digestive enzyme activity, nutrient transporter expression as they are very expensive and this could be another reason for studying more parameters such as antibody titers against Newcastle Disease Virus (NDV) and Avian Influenza Virus-Type 9 (AIV-H9) in a trail to cover the possible mechanism under investigation.
The data concerning the use of chromium propionate on antibody titers against Newcastle Disease Virus (NDV) and Avian Influenza Virus-Type 9 (AIV-H9) of broilers are scanty, and the previous studies on the use of chromium propionate in broiler rations resulted in contradictory conclusions. Therefore, the current research study has been designed to evaluate the dietary effects of chromium propionate on growth performance, blood biochemistry of broilers (i.e. glucose, liver enzymes ALT, AST, ALP, cholesterol, LDL, HDL and triglycerides) and antibody titers against NDV and AIV-H9.
There are lots of literature that have already looked into the parameters as the authors mentioned in the manuscript. Its a known knowledge that Chromium increases glucose uptake and may reduce triglyceride levels. So it would have been good to look at body composition of these birds to validate those rather than repeating the same again.
Many thanks for the valuable comments and great efforts of all reviewers to improve our manuscripts
Best regards,
Corresponding author
Round 2
Reviewer 1 Report
Dear authors,
thank you very much for the effort in rewriting the manuscript. Still, I really recommend not to repeat the results in detail in the discussion. You might delete many sentences in the discussion.
Kind regards
Author Response
Thank you very much for your great effort and valuable time you spent in reviewing our paper and for giving us the chance to clarify our work. We removed many sentences (unnecessary sentences) in the discussion to improve the paper. We wish these changes would improve our paper upon your helpful suggestions.
Many thanks for the valuable comments and great efforts of all reviewers to improve our manuscripts
Reviewer 2 Report
I recommend to accept it
Author Response
Thank you very much for your great effort and valuable time you spent in reviewing our paper. Thanks again for your supportive comment